# COVID-19 Vaccination Drive in a Low-Volume Primary Care Clinic: Challenges & Lessons Learned in Using Homegrown Self-Scheduling Web-Based Mobile Platforms

**DOI:** 10.3390/vaccines10071072

**Published:** 2022-07-03

**Authors:** Reita N. Agarwal, Rajesh Aggarwal, Pridhviraj Nandarapu, Hersheth Aggarwal, Ashmit Verma, Absarul Haque, Manish K. Tripathi

**Affiliations:** 1Department of Internal Medicine, VA Hospital, Memphis, TN 37132, USA; 2Department of Information Systems and Analytics, Middle Tennessee State University, Murfreesboro, TN 37132, USA; rajesh.aggarwal@mtsu.edu; 3Care Eco Healthcare, Brentwood, TN 37027, USA; raj@careeco.net; 4Health Science Center, College of Medicine, The University of Tennessee, Memphis, TN 38104, USA; haggarwal@uthsc.edu; 5DivyaSampark iHUB Roorkee for Devices Material and Technology Foundation, Indian Institute of Technology Roorkee, Roorkee 247667, India; ashmitverma1998@gmail.com; 6King Fahd Medical Research Center, Department of Medical Laboratory Science, Faculty of Applied Medical Sciences, King Abdulaziz University, Jeddah 21589, Saudi Arabia; mhaque@kau.edu.sa; 7South Texas Center of Excellence in Cancer Research, Department of Immunology and Microbiology, School of Medicine, University of Texas Rio Grande Valley, McAllen, TX 78504, USA

**Keywords:** COVID-19, vaccination, accessibility, web-based platforms, self-scheduling

## Abstract

**Background:** The whole of humanity has suffered dire consequences related to the novel coronavirus disease 2019 (COVID-19). Vaccination of the world base population is considered the most promising and challenging approach to achieving herd immunity. As healthcare organizations took on the extensive task of vaccinating the entire U.S. population, digital health companies expanded their automated health platforms in order to help ease the administrative burdens of mass inoculation. Although some software companies offer free applications to large organizations, there are prohibitive costs for small clinics such as the Good Health Associates Clinic (GHAC) for integrating and implementing new self-scheduling software into our e-Clinical Works (ECW) Electronic Health Record (EHR). These cost burdens resulted in a search that extended beyond existing technology, and in investing in new solutions to make it easier, more efficient, more cost-effective, and more scalable. **Objective**: In comparison to commercial entities, primary care clinics (PCCs) have the advantage of engaging the population for vaccination through personalized continuity of clinical care due to good rapport between their patients and the PCC team. In order to support the overall national campaign to prevent COVID-19 infections and restore public health, the GHAC wanted to make COVID-19 vaccination accessible to its patients and to the communities it serves. We aimed to achieve a coordinated COVID-19 vaccination drive in our community through our small primary care clinic by developing and using an easily implementable, cost-effective self-registration and scheduling web-based mobile platform, using the principle of “C.D.S. Five Rights”. **Results:** Overall, the Moderna vaccination drive using our developed self-registration and scheduling web portal and SMS messaging mobile platform improved vaccination uptake (51%) compared to overall vaccination uptake in our town, county (36%), and state (39%) during April–July 2021. **Conclusions:** Based on our experience during this COVID-19 vaccination drive, we conclude that PCCs have significant leverage as “invaluable warriors”, along with government and media education available, to engage patients for vaccination uptake; this leads to national preventive health spread in our population, and reduces expenses related to acute illness and hospitalization. In terms of cost-effectiveness, small PCCs are worthy of government-sponsored funding and incentives, including mandating EHR vendors to provide free (or minimal fee) software for patient self-registration and scheduling, in order to improve vaccination drive access. Hence, improved access to personalized informative continuity of clinical care in the PCC setting is a “critical link” in accelerating similar cost-effective campaigns in patient vaccine uptake.

## 1. Introduction

The whole of humanity has suffered dire consequences of the novel coronavirus disease 2019 (COVID-19). Worldwide, the pandemic declared in March 2020 reached a catastrophic scale, overwhelming health care systems and causing substantial loss of life. The case fatality rate ranged from 2% to 3% [1,2]. While this has been challenging, there have also been critical scientific discoveries and speedy development of effective vaccines against COVID-19. Vaccination of the world base population has been considered promising yet very challenging. It required the safe and effective delivery of millions of vaccine vials in the shortest times possible, while avoiding health inequalities. After research, manufacturing, and distribution challenges came the challenge of engaging and administering the vaccine to the population in order to achieve herd immunity [3]. The task of inoculating about 7 billion eligible citizens 12 years or older was immense. Just as scientific and pharmacological companies had to create new infrastructure, similarly, clinics administering COVID-19 vaccines had to create new facilities, tools, and procedures for the delivery of vaccines to willing recipients [4,5].

As the WHO reports, over 530 million positive cases of COVID-19 cause more than 6.3 million deaths, with over 1.1 billion vaccine doses administered worldwide [6]. Currently, there are many vaccines available in the market, such as Moderna/Pfizer (mRNA based), Zycov-D (D.A.N. based), NVX-CoV2373 (protein-based), AstraZeneca (viral vector-based), etc., which show promising results and efficacies. However, there are some challenges, debates, and issues associated with vaccination as well, such as defining booster doses; re-infection due to the emergence of new variants of the virus, herd immunity development; clinical trials related to elderly persons, pregnant women, and small children; vaccine hesitancy, diplomacy, and availability of vaccines to the standard population worldwide [7]. In addition, researchers worldwide are concerned about the decreasing acceptance rate of vaccination. A study found that in a vaccination drive conducted on a group of 2953 health care workers, only 69% of the participants agreed to accept a vaccine [8]. Another study found that 10% of primary care physicians (P.C.P.s) do not want to take the Pfizer and Moderna vaccines, and 32% do not want to take the Johnson & Johnson vaccine [9]. It was suggested that promoting the sharing of COVID-19 vaccine personal narratives on social media, showing the reason for COVID-19 vaccine hesitancy, and emphasizing freedom from fear once vaccinated, could collectively be effective means in reducing COVID-19 vaccine hesitancy among the population [10]. Moderna’s website also has all updates about their vaccine in regards to adverse events or age changes and acceptance, which are constantly changing [11].

In the U.S.A, vaccinations began in mid-December 2020 for front-line healthcare workers. Commercial pharmacies such as Walgreens, Walmart, and CVS were recruited to make COVID-19 vaccination easily accessible for communities through government-sponsored mass vaccination events. As healthcare organizations took on the extensive task of vaccinating the entire U.S. population, digital health companies, namely Notable Health, Zocdoc, Relating, Nimber, Well Health, Kyruus, MEDITECH health, NextGen EHR, Experian Health, MEND, etc., expanded their automated health platforms to help ease the administrative burdens of mass inoculation. Some offered free applications to large organizations. However, there were prohibitive costs for small clinics such as ours in integrating and implementing them into our e-Clinical Works (ECW) Electronic Health Record (EHR). These cost barriers forced us to look beyond what we had, and resulted in the investment in new solutions to make it easier to carry out mass inoculation more efficiently, more cost-effectively, and in a more scalable way.

Primary Care Clinics (PCCs) are considered critical links to achieving high rates of vaccinations because these facilities have earned patients’ trust over years of providing health services [12]. Therefore, in order to support the overall national campaign in preventing COVID-19 infections and restoring public health, we wanted to engage patients and make COVID-19 vaccination accessible to our clinic patients and the communities we serve. Therefore, we felt encouraged to consider homegrown EHR application when we read about the Office of Civil Rights (OCR) issuing notice on 19 January 2021, stating that it will not impose penalties for HIPAA non-compliance in connection with a covered entity healthcare provider’s or business associate’s good faith use of online or web-based scheduling applications (WBSAs) for the scheduling of appointments for COVID-19 vaccinations during the public health emergency [13].

We aimed to achieve a coordinated COVID-19 vaccination drive in our community through our small primary care clinic by developing and using an easily implementable, cost-effective self-registration and scheduling web-based mobile platform.

## 2. Materials and Methods

### 2.1. Facility

The Good health Associates Clinic PLLC (GHAC) is a relatively low-volume primary care clinic in Murfreesboro, Tennessee, with an active patient panel size of 2008. Opt-in for contact via mobile telephone numbers in March 2021 for the COVID-19 vaccination drive was planned. The clinic had used ECW EHR since August 2008, since it was first opened. Due to COVID-19, there was a skeleton crew that consisted of a clinic manager, two medical assistants (MA), nursing staff, and one full-time receptionist assisting 2.25 providers. In addition, the clinic had an intermittent third MA and a receptionist whenever possible or available.

The vaccination drive for prevention against COVID-19 at the GHAC began in April 2021. Moderna (MRNA) vaccine from the Tennessee State Government was used for this drive. This drive resulted in the vaccination of more than 1100 patients in April–July 2021. This article is about lessons learned during the vaccination drive at the GHAC.

### 2.2. Pre-Clinical Planning Activities for Identifying Barriers to COVID-19 Vaccination Drive

Complexities during the COVID-19 vaccination process arose from several sources. While planning the COVID-19 vaccine clinic, the GHAC assessed their needs to prepare through interaction with clinical staff, and reviewing the federal, state, and local guidelines [5]. Required COVID-19 sanitation supplies and measures were already secured from GHAC regular patient care. COVID-19 vaccine storage and administration training of staff were through VPDIP (Tennessee Vaccine-Preventable Diseases and Immunization Program). The workflow protocols were assessed and addressed to develop the vaccination clinic workflow. Clinical staff were involved in finalizing and updating the workflow of COVID-19 vaccination. We reviewed barriers unique to eligible patients as per the CDC and local guidelines special to the COVID-19 Moderna vaccine, our GHAC building/ facility layout, staff, and ECW EHR to meet our objective of vaccine coverage/uptake of our panel and the surrounding community.

#### 2.2.1. Unique Issues Related to COVID-19 Moderna Vaccine

The GHAC enrolled with the state of Tennessee for a vaccination drive in late 2020 as a volunteer to the state of Tennessee to do a vaccination drive for the Moderna vaccine, in order to minimize staff burden and expenses for new resources to handle it. The Moderna vaccine, a two-dose vaccine for those aged 18 and older, was chosen for our facility. It could be delivered and stored frozen between −58 F and 5 F. Unpunctured vials could be held for 30 days in a regular refrigerator from 36 F to 46 F and punctured at 36 F–77 F for 12 h. Each vial contained 10 doses. There is a recommended time of 26 days between administering the first and second doses.

Reviewing Moderna vaccine information, we did not have to purchase any new equipment for storage. Preparation was easy to keep the required vials at room temperature for 2 h before the vaccine clinic began. We had to make sure that we had a capacity setup for using one whole vial of 10 doses in order to minimize wastage and provide reminders for the second dose of the vaccine [11].

We also devised a workflow to minimize vaccine wastage for clinic appointments. At our clinic, we observed a nearly 20% no-show rate. We used a binomial function to determine the optimal number of appointments to schedule in order to minimize wastage of vaccines, while maintaining at least 95% confidence that the registered person will show up to receive the vaccine within batches of 10 doses. The clinic scheduled 10–20% (3–5 above slot capacity) more patients on a regular schedule (see Table 1 for probabilities generated by the binomial distribution function for optimal utilization of the vaccine). At the start of vaccination clinic time, if patients were not registered in batches of 10, we would initiate a backup plan of calling our community volunteers to send their acquaintances if we were short of 1–3 doses from a batch of 10, or occasionally reschedule 1–2 patients for the next day if they cooperated.

#### 2.2.2. Unique Issues Related to Eligible Patients Based on CDC and Local Government Guidelines

CDC and local government had established ongoing updated eligibility of population for vaccines based on multiple criteria. In addition, information about the vaccine was updated based on new research data. It was decided that this information would be part of the EHR-generated algorithm, in order to generate a list of eligible patients for scheduling. The software had to provide patients with their preferred choices in scheduling vaccines as well.

#### 2.2.3. Unique Issues Related to Staffing Barriers: Preventing Sickness and Burnout

Due to COVID-19, we operated the clinic with a skeleton crew, as mentioned above. We had to protect staff from sickness and prevent burnout from additional workload while maintaining our new and regular follow-up patient appointments. We required a solution to reduce the staff documentation burden and avoid burnout while serving the population. As a result of the COVID-19 pandemic, hiring more staff was very difficult and costly due to the nursing shortage nationwide. A strategy had to be devised to provide customer service and balance the cost of appropriate staffing needs while we were in the middle of reduced revenue, in order to meet the expenses of running our clinic.

#### 2.2.4. Unique Issues Related to Clinic Facility Layout: Flow of Patients to Reduce Exposure

The following is a list of the main issues identified:Modified parking lot entry/exit: For smoother traffic flow, one-way entry and exit from our parking lot was required. The exit should be closer to parking spaces nearer to the nurse vaccination area.Negative pressure nurse vaccination room: For safety and to reduce crowding in the lobby, a nurse vaccination room with a second door opened towards the parking lot’s reserved spaces, serving as “curbside vaccination” access after reception verification was performed. This room should have a negative pressure air conditioning setup not connected with the rest of the clinic.

#### 2.2.5. Unique Issues Related to Our ECW Health Record System: Automate Scheduling of Clinic and Community Population

Unique issues related to our ECW EHR as barriers to performing a vaccination drive were identified as follows:Our ECW EHR could only send messages via the patient portal to patient emails and mobile text, if opted in. However, patients can be called or texted otherwise for their clinic care.In our panel, patient portal opt-in was only about 30% of our active patient panel.Patient portal messaging did not have the capability of self-scheduling or self-registering appointment slots in any staff clinic. It could only send messages to request an appointment or send reminders for appointments; self-scheduling or rescheduling could not be done instantly. With this setup, we could not offer self-scheduling access for vaccination to our panel, nor to non-clinic patients in our community.Due to the policies and governance issues of ECW, their staff customer service refused to provide any assistance with integration, or otherwise, for self-scheduling COVID-19 vaccination, in the case that we developed an API or software for our clinic. Instead, ECW gave us a quotation of $15,000 as the cost for integrating an API from other vendors’ developed software in the market.ECW already had access to TennIIS to push vaccine data; however, the state had not yet set up for COVID-19-vaccine automated loading and pushing of data during March–April 2021 when we planned to initiate the vaccination drive.

### 2.3. Solutions to Overcome Barriers

In order to run a successful COVID-19 vaccination drive, we identified the need for a staff- and patient-friendly solution with a cost-effective vendor relationship that could scale with our staff and our patient needs. Evidence suggested that there is a moderately positive impact of PEHR (Patient portal Electronic Health Record) access in increasing vaccine uptake, based on data for influenza and pneumococcal vaccines, diabetic patients, and childhood immunization [11,14,15,16,17,18,19]. Based on our ECW patient-portal issues mentioned earlier, and on literature concerning various modes of immunization trials, we concluded against using this model.

In order to support the overall national campaign in preventing COVID-19 infections and restoring public health, it was planned to make COVID-19 vaccination accessible to GHAC patients and to the communities it serves. Patient self-scheduling offered healthcare facilities a way to lift some barriers to vaccination accessibility, empowering patients to self-select and find appointments themselves, freeing up phone lines and staff workload. Therefore, the GHAC approached an in-house IT collaborating team, Care Eco Corporation (CEC), to help develop and implement a cost-effective and efficient “Patient Self-registration & Scheduling” application, in order to circumvent potential barriers. CEC helped meet the need within two weeks, while vaccines were en route from the state of Tennessee in March 2021.

With the homegrown “Patient Self-registration & Scheduling” solution, the GHAC attempted to make it simpler for patients of the GHAC and the community it serves to select dates, locations, and times that worked for them from our GHAC website linked to the mobile platform. No back-and-forth phone calls and no patient portal were necessary.

### 2.4. Goal/Objectives/Outcomes

The GHAC’s overarching goal was to improve access and rates of COVID-19 vaccination among its community’s eligible population. The GHAC, a relatively small primary care clinic; aimed to achieve a coordinated COVID-19 vaccination drive in its community by developing and using easily implementable, cost-effective self-registration and scheduling online.

We are based on a mobile platform. The primary outcome of interest was to measure (as percentages) vaccination in our clinic population and surrounding community: vaccine coverage or vaccination uptake. Our secondary outcome aim was to improve access to healthcare for vaccination while preventing an excessive increase in workload burden on clinical staff, or financial pressure on the clinic’s operational budget.

### 2.5. Theoretical Model of “C.D.S. Five Rights”

A framework that proved helpful for achieving success in our vaccination effort is the “C.D.S. Five Rights” approach. A GHAC physician with a clinical informatics background collaborated with CEC to develop a self-scheduling (registration and scheduling appointment) web-based mobile application that integrated with our ECW scheduling application. The architecture of this software for the flow of information is detailed in Figure 1a,b.

The GHAC devised SMS (Short Message Services) messages that were sent to our clinical population, in order to set up appointments using SMS message links. The SMS message enabled patients to verify/update/provide personal information and demographics, insurance, and their Tennessee driver’s license numbers for check-in verification. They also had to choose the eligibility criteria, fill out the consent form and screening form for the COVID-19 vaccine, and acknowledge contraindications and precautions related to vaccine information and emergency use authorization. It also provided educational information about vaccines and reporting of adverse events. In order to develop the CEC scheduler SMS, we used a rule-based decision tree intelligent algorithm that adopted the clinical workflow for scheduling.

The core functionality of this schedular was to generate an appointment block based on the rule engine that consisted of several parameters as follows:Purpose of visit;Number of treatment rooms;Number of staff clinic schedules;Number of slots in each clinic;Time window (specific day/ days of a week, specific hours in a day);Appointment duration;Guidelines in selecting patient panel (based on state-/CDC-/clinic-specific guidelines);Moderna vaccine-related adjustment in schedule.

Based on the algorithm, selected patients were sent SMSs, as shown below, containing a secure web link—staff initially set up one clinic from 2 pm to 4 pm with 5-min slots daily. In order to reduce wastage of vaccines with 10 doses per vial, the team vaccinated in batches of 10. As a result, the GHAC overbooked 20% of their panel and had a backup waitlist to come in as walk-ins between 3:30 pm–4:00 pm. Staff updated their self-registration and scheduling algorithm slots weekly based on CDC eligibility updates, vaccine demand, staff available, and numbers of vaccine vials to be used (10 doses per vial) in order to prevent wastage. In order to provide access to the non-clinic population, on the GHAC website a COVID-19 vaccine scheduling link was set up with SMS capability on the mobile platform for easy access. This link was also provided on the Government “vaccine Finder” website for easy scheduling.

Based on “C.D.S. Five Rights” of SMS alerts to patients, the following are supporting comments:The correct information: Information provided was aligned to CDC initiatives, patient preference of appointment slot, clinic documentation, and billing information pre-entered for the clinic staff workflow. We also embedded vaccine related information and forms for patients as they set up appointments.To the right person: All stakeholders’ needs were considered: CDC and local government, patient, check-in staff, nurses, and billing. Information about appointments and vaccines was visible to all shareholders in order to facilitate efficient patient and staff workflow. The benefit of a choice of appointment in patient’s data is that receptionist did not have to enter it, only verify the patient’s relevant clinical information to prevent adverse events. In addition, the nurse could verify and do consent verification.In the proper C.D.C. intervention format: an SMS alert message was sent with embedded information to the patient for engagement for an appointment with choice of clinic date and time.Two channels were used through the right channel: through the internet web portal and mobile platform, both clinic and non-clinic patients could access mobile platform scheduling and receive SMSs for engagement and confirmation. While GHAC patients were engaged through mass SMS reminders and scheduling immediately at their convenience with a confirmation response, our non-clinic patients signed up through a web link with the mobile platform which provided the same services.At the right time in the workflow: Patients had access from their mobile phones to schedule appointment for times they preferred. A reminder on the event day was also sent to the patient with the capability of confirming or rescheduling. In addition, clinic staff reception and nurses had IPADs to access appointments with patient data in order to plan vaccine doses needed for that day. Vaccine and scheduling updates were done every week or sooner if the state criteria changed. The system also verified when the patient arrived for check-in, and the vaccine that was received.

### 2.6. Workflow during Vaccination Clinic

The physician with an applied clinical informatics background, the clinic manager, and Care Eco Corporation (CEC) staff collaboratively engineered and optimized workflow. All vaccinations were done by appointment, and were scheduled via an online GHAC web link with the mobile platform. Our clinic’s patients were sent SMS messages to schedule for vaccinations. Walk-ins were informed to visit our website with their smartphone in order to set up appointments. As part of process optimization, clinical workflow with staff responsibilities during vaccination is shown in Figure 2, which also shows the architecture of self-scheduling software applications.

The steps in information and workflow to engage as well as set up vaccination appointments using the CEC platform are as follows:The GHAC’s existing patient panel demographics, including phone numbers, were uploaded into the CEC platform. The CEC engine selects qualified patients based on state guidelines.All patients in the clinic’s database are sent SMSs regarding vaccine availability. They can also access this information through website links, just as non-clinic patients can.Patient interface: Each qualified patient received a secure and unique portal link with their prepopulated demographics data. The person could enter/update personal information, including their medical insurance information, and driver’s license number. Then, the person clicked on answer choices for eligibility criteria, the vaccination screening questionnaire, and consent.Software provided available time slots. The person selected a time slot. The system blocked for same-day scheduling unless there were unfilled slots. This helped the patient quickly schedule appointments with the touch of a button.Once a patient scheduled an appointment, the database scheduled and sent a text message of confirmation.Another SMS with an option to reschedule/cancel and containing vaccine-related documents was delivered on the day of the appointment.At clinic reception interface—patients presented their Tennessee driver’s license or a form of government I.D. and health insurance card if insured. They filled in/verified the health screening questionnaire.CEC workflow provided a real-time view of upcoming appointments and changes to existing appointments to our staff in the CEC platform and ECW schedule.As soon as a patient received the first dose and staff checked out, a follow-up message with a pre-calculated date range was delivered via SMS to the patient regarding their second dosage appointment.Follow-up reminders were sent 72 h before the appointment via SMS as well as on the day of the appointment. We had to ensure that follow-up reminders did not fall on weekends when the clinic closed.

The steps in the clinical workflow of patient and staff activities during vaccination are as follows:Any patient, especially non-clinic patients, accessed the healthcare organization’s website via a mobile device or computer, clicked the link to “get an appointment,” and selected “COVID-19 vaccine.”All patients in the clinic’s database were sent SMSs regarding vaccine availability.The person entered/verified personal information, including medical insurance, driver’s license number (or other identification information), in addition to completing the vaccination screening questionnaire.Software provided available time slots.The person selected a time slot.The system blocked same-day schedules unless there were unfilled slots.Database chedules and text messages to the patient for confirmation.The clinic could see each day’s appointments in their ECW EHR portal through the CEC system.At clinic reception—the patient presented a Tennessee driver’s license or a form of government I.D. and a health insurance card if insured.The information was filled/verified in the health screening questionnaire.The patient signed the printed online/on-site prefilled consent forms.The receptionist filled out the proof of vaccination card and gave it to the nurse, along with the consent form and vaccine screening questionnaire.The vaccination nurse verified the required screening questions.The nurse administered the shot. If a patient did not meet eligibility because of immune compromise or other issues, they were requested to get approved from PCP or to use our on-site provider for appointment.The patient is provided a filled immunization card with vaccine vial information and the date of first vaccine or the second vaccine and, if needed, was set up a reminder for the second dose of vaccine.Patient/receiver of vaccine waited 30 min in the lobby, nurse room, waiting room, or car parking lot near the vaccination room, and was verified for any adverse event.Nursing staff pushed data of patients administered COVID-19 vaccines at the end of the day (24 h to 72 h max) to the VAMS and TennIIS dashboards after reconciling in ECW until the integration of new software.Nursing staff or managers pushed data for reimbursement to state government and insurance agencies.

### 2.7. Post-Vaccination Clinic Activities

Planning and implementation of the vaccination drive is a continuous process. At the end of each day’s COVID-19 vaccination session, the team convened with the clinic manager and team leader on-site in order to identify successes, challenges, and barriers encountered during operations; procedures were refined or adjusted as needed. In addition, several iterations of the following were conducted in order to improve readiness as we refined workflow daily:Testing workflow, protocols, workload, and procedures;Training of volunteers and staff;Fine-tuning patient traffic flow in order to reduce crowding at the reception check-in station.

## 3. Results

Our COVID-19 vaccination drive’s primary outcome of interest was to improve the measure (percentage) of COVID-19 vaccination uptake in our clinic population and surrounding community, in comparison to Rutherford County and the Tennessee state census. The GHAC achieved better results than Rutherford County or Tennessee State census on COVID-19 vaccination uptake.

With a total of 5895 SMS messages sent to 2008 patients in our panel, the GHAC achieved 51% of the panel becoming fully vaccinated (2 doses) compared to August 01, 2021, Rutherford County achieving 36.5%, and Tennessee state 39% [20,21]. Total vaccinated patients at our site were 1103, with 993 from our panel and 110 from outside our panel (Figure 3). We noted that 2.9% opted not to get vaccinated at all. We are still updating data for the rest of the panel who did not reply to the SMS message, and for non-patient portal signup patients.

Reviewing gender (Figure 4a) and ethnicity (Figure 4b) data, we noted that vaccination uptake in males (60%) was slightly higher than in females (54%), in comparison with the state of Tennessee which also reported higher vaccination uptake for males 54% than females 45%. At the GHAC, female uptake was significantly higher than state data, while male data was close to state census. Two significantly higher peaks observed in females 50–60 years followed by 30–40 years versus male age data, with similar height peaks in all other age groups, explain the differential among female gender vaccination uptake (Figure 3). However, the reason is not apparent. There are fewer vaccinated individuals aged >70 for both genders, since most of them were already vaccinated before we started our drive at the top of the eligibility list. Regarding race data, it is difficult to make any observations regarding differences in vaccination rates based on race and ethnicity, since most patients opted out of answering related questions.

Our secondary outcome aim was to improve access to healthcare for vaccination while preventing an excessive increase in workload on our staff, or in financial costs on our clinic’s operational budget. The GHAC achieved improved vaccine access as seen by enhanced vaccination uptake rates compared to Rutherford County and Tennessee state during this project’s 4 months (April–July 2021). In addition, our vaccine wastage was only 5%. However, the GHAC had to hire a person for 30 h per week in order to help with the additional documentation required and to assist at the reception desk. Strictly in terms of monetary rewards, the clinic lost money in vaccination while initially awaiting government and insurance reimbursement. Still, GHAC staff felt honored to be part of fighting the pandemic and contributing to normalcy.

The Good Health Clinic attributes success in their vaccination drive to the teamwork of GHCA staff and the CEC team dedicated to our patients and community. They worked towards engaging our patient base to become vaccinated, nudged by an SMS system that sent meaningful intermittent text messages. Ease of scheduling, data input, and appointment confirmation through a self-scheduling, web-based mobile platform further encouraged people to sign up willingly. Above all, good rapport between patients and their primary care providers gave candidates the confidence to go ahead with their decision to get vaccinated just in time, and at their convenience.

## 4. Discussion: Benefits and Challenges

A COVID-19 vaccination drive was performed with some challenges, but adequately addressed barriers that faced a relatively successful outcome. The performance measures and tasks associated with the activity were completed in a manner that achieved our primary objectives of improving access and vaccination uptake at 51%, compared to Rutherford County at 11% and Tennessee state at 17% in April 2021 when we started planning the drive, to 36% and 39%, respectively, on 1 August 2021, when we reviewed these data points. It did not negatively impact the performance of other activities, such as regular patient follow-up appointments of our panel. The performance of this activity did not contribute to additional health or safety risks for the public or GHAC workers, and it was conducted in accordance with applicable guidance and plans aligned with CDC recommendations. However, daily meetings post-vaccination in clinic sessions identified opportunities to further enhance effectiveness and efficiency.

### 4.1. Major Strengths

Primary care clinics (PCCs) have the advantage of engaging populations toward receiving vaccination through their personalized care and as a result of the good rapport patients have with their PCC team, compared to commercial pharmacies such as Walgreens, CVS, Walmart, and walk-in/urgent care clinics. Such entities provide easy access; however, since they do not have a complete clinical history of patients or a relationship of trust, they may not be able to engage reluctant patients sufficiently for vaccination uptake. For patients, the PCC is a trusted venue due to relationships that have been developed over time. Also, for continuity of care, the PCC team can clarify lingering questions related to vaccines, and provide quick access to personalized recommendations if adverse reaction-related issues occur and require treatment. KFF COVID-19 vaccine monitor survey (6) provided supporting evidence that if COVID-19 vaccine was available in their own doctor’s office, 75% (highest) of people say they would “very likely” get the vaccine there, compared to 61% for local pharmacies, and much lower percentages for other locations such as hospitals, community health clinics, workplaces, etc. (Figure 5). The survey further showed that when forced to take a COVID-19 vaccine, again, 38% (highest) would prefer going to their doctor’s office if possible. Hence, the PCC is an essential contributor to vaccination uptake in the community and to preventive health spread.

Major strengths identified during this exercise are as follows:Patient: During the early phases of vaccination in March 2021, there was some distrust about the vaccine. The population was unsure about their eligibility, lacked information about relevant vaccine literature, and about when or where they could get vaccinated. Patient self-scheduling empowered people to self-select appointments and receive confirmation just in time. They also received ready access to their eligibility and other vaccine-related information. The patient portal of ECW did not have that level of a quick solution.Clinic staff: The reception check-in and nursing staff had the capability of pre-planning and experienced some reduced workload burden. They could see each day’s appointments in their EHR portal and CEC interface to facilitate planning vaccination vial use with immunization cards to be set up. In addition, the system had automated messaging for arranging second doses. Staff also did not have to enter patient data or fill out their vaccine related consent forms and clinical screening forms; instead, this information was prefilled by patients, and had only to be verified.Clinic facility: The CEC self-scheduling application helped engage with our patient panel and update their information. We also had some new patients join our clinic. The CEC application also served to preregister new clinic patients who had not previously been seen as patients, and self-select regular follow-up appointments for our panel patients.

### 4.2. Major Challenges and Opportunities for Improvement in Workflow Performance

Throughout the vaccination drive, several opportunities for improvement in workflow were identified. In addition, significant challenges and lessons learned were best understood during the workflow meetings and follow-up iterations.

The CEC self-scheduling mobile platform initially presented some challenges for patients and staff. We ran our first vaccination clinic session with patients whom we knew personally. As per instructions, they scheduled appointments using their smartphones while on-site. At that time, all staff, including the CEC IT person, were also on-site. It was noted that patients experienced difficulty entering their birth year fields with the pop-up calendar on the mobile platform but not with the web-based GHAC online. CEC fixed that issue within 2 days, providing the date field as a drop-down scrolling choice of numbers 0–9 for each day, month, and year.

During our first two days, we noted that the data of vaccines administered to patients were not transmitting to the VAMS (Vaccine Administration and Management System) and TennIIS (Tennessee Immunization Information System) dashboards automatically from our CEC interface. The CEC interface was integrated with ECW scheduling and patient demographics, but not with the clinical data of patients. Hence, the nursing staff were required to complete additional documentation of patients administered COVID-19 vaccines at the end of the day (24 h to 72 h max) for the VAMS and TennIIS dashboards after first reconciling in ECW. The software developed by the federal government did not work as desired. Clinics had to receive assistance from different software vendors. Some providing software companies were quoting as high as $20,000 for the software, and an additional $1 per transaction. This was cost-prohibitive for a small clinic because the shelf-life of this vaccination project is short-term. ECW refused to allow CEC to integrate further with ECW, quoting a cost of about $15,000 instead of recruiting us for API development and integrating with our practice. Our team did not receive assistance from the respective staff of VAMS and TennIIS in automating updates of patient data directly onto these state portals, since they were still in their early phases of planning. This resulted in an increased workload for our nursing staff. We were unable to hire more nursing staff due to shortages, and consequently found an alternative part-time staff member to enter the data into these state portals via reconciliation in ECW. Automation and integration of these data with the CEC interface will help our nursing staff tremendously. We are still in communication with them.

To our surprise, initially, the numbers of walk-ins and phone calls increased for COVID vaccine appointments. During our analysis, we noted that the reason for this was because the GHAC self-scheduling appointment portal was not integrated on vaccinefinder.org to redirect patients to our website for self-scheduling. The site was only showing our facility name, address, and phone number. We reached out to the respective government to address this issue. They were able to add our website address but have not yet linked to our self-scheduling portal. Usage of the website helped in reducing some phone calls and walk-in crowds. However, this also prevented the non-clinic population from easily self-scheduling appointments. We are also updating our automated message regarding the COVID-19 vaccination self-scheduling portal.

In order to minimize vaccine wastage, we faced the challenge of administering vaccines in batches of 10. When we initially began requesting patients to reschedule as the end of the day approached, some patients expressed their unhappiness and provided negative feedback. We then devised a method to call our community liaison to send candidates to fill deficits in the batches of 10 vaccinations; we also added five slots for overbooking on the same day in order to achieve this. The clinic manager handled these tasks in order for remaining staff in the workflow not to become overburdened.

During the first week, there was patient crowding at check-in even though we provided automated check-in with an iPad on a stand, in addition to the online self-scheduling portal. The number of patients coming to our clinic doubled with the addition of vaccination clinic slots from 2–4 pm. Patient slots were 5 min each. Some people came as walk-ins, and this became our bottleneck in the first week of the drive. Our manager and ECW IT personnel had to assist in order to prevent overcrowding and redirectionof the workflow at check-in. Patients checked in on paper and had to wait in their cars to be called in if there were already more than four people. IT personnel or the second check-in receptionist used an iPad to check visitors in and distribute packets of printed vaccination forms for the next step. Walk-in patients were sent an SMS message to schedule an appointment for the same day as overbooking (five maximum), in order to stay close to maintaining doses in batches of 10. In order to resolve overcrowding at check-in reception for COVID-vaccine walk-ins, we also displayed a flier at the entrance and in the lobby with information about our website self-scheduling portal to be used on their cell phones. We also used check-in instructional fliers for patients who arrived for streamlined check-in on the iPad in the lobby; they then waited in their cars until they were sent a message to come in for the next steps in receiving their vaccinations.

The traffic of patients in our parking lot increased from the previous 25–30 patients per day by an additional 30 patients scheduled for appointments as well as walk-ins. Our existing 42 parking spaces, in addition to the entry and exit, had to be defined and labeled. We reserved two slots for vaccine administration at the nurse station, opening towards the parking lot. We designated entry and exit areas for one-way traffic flow. Our staff parked on the grass behind our clinic in order to keep more parking slots available. During the first few weeks, our maintenance person monitored and directed traffic flow in the parking lot.

## 5. Conclusions

In general, our COVID-19 vaccination drive was both efficient and cost-effective in achieving our goal of improving vaccination uptake and preventing an excessive increase in clinical staff workload, while balancing the expensebenefit economics of the clinic that resulted from the vaccination drive.

Overall, the vaccination drive using self-registration and scheduling web portals and SMS messaging improved patient engagement and access, leading to higher vaccination uptake compared to overall vaccination uptakes in our town, county, and state. Our vaccine penetrance and patient outreach would have had a more significant impact in the community if our GHAC appointment portal had been integrated into the government website vaccinefinder.org during the early months of the vaccination drive, as the site was only showing our information and location.

Our clinical staff, in general, were satisfied and not overburdened except during some challenges encountered in the first week for workflow issues. Our ongoing unresolved manual vaccine data entry into VAMS and TennIIS awaiting automation from the Government of Tennessee assigned department to resolve has resulted in some unexpected documentation burdens on our nursing staff. Otherwise, the team expressed that patient management improved, and that service was provided with no excessive workload.

Based on our experience during this COVID-19 vaccination drive, we conclude that any PCC (small or large) has significant leverage as “invaluable warriors”, along with government and media education, in engaging patients for vaccination uptake, all of which lead to national preventive health spread in our population in addition to reduced expenses related to acute illness and hospitalization. In terms of cost-effectiveness, small PCCs are worthy of government-sponsored funding and incentives, including mandating EHR vendors to provide free (or minimal fee) software for such patient self-registration and scheduling in order to improve vaccination drive engagement and access. Throughout the country, the way PCCs operate is ideally suited to implementing similar cost-effective, efficient SMS-based reminders and self-scheduling campaigns regarding COVID-19 vaccine information and inoculation in their communities. These clinics can become invaluable “warriors” in fighting vaccine misinformation and in engaging the population through personalized clinical care to increase vaccination uptake. Hence, improving access to personalized informative continuity of clinical care in the PCC setting is a “critical link” in accelerating similar cost-effective campaigns of patient vaccine uptake.

## Figures and Tables

**Figure 1 vaccines-10-01072-f001:**
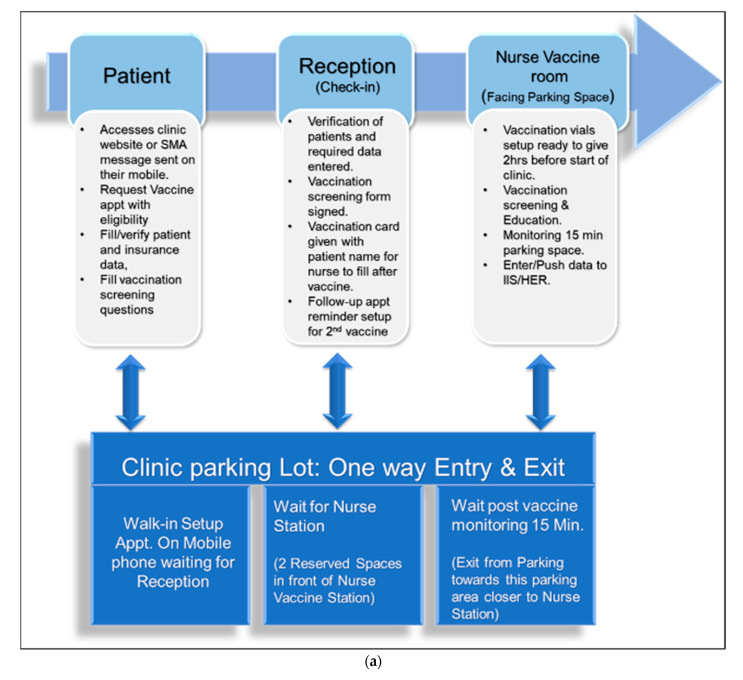
(**a**) Clinical workflow of patients and staff. (**b**) Architecture of the self-scheduling app.

**Figure 2 vaccines-10-01072-f002:**
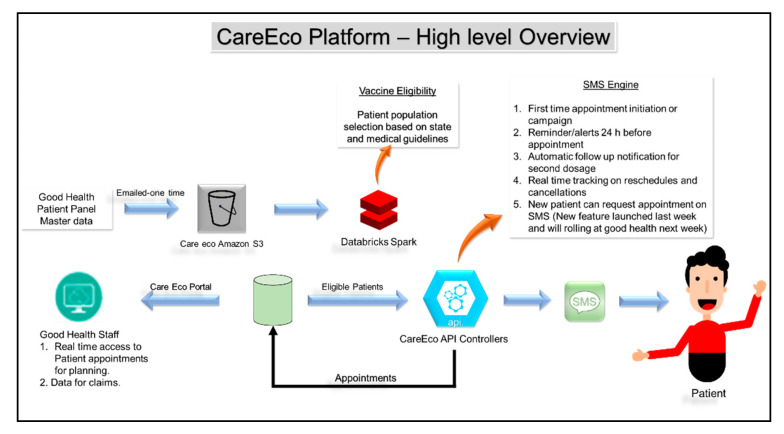
The flow of information for the Good Health Clinic using the scheduling app.

**Figure 3 vaccines-10-01072-f003:**
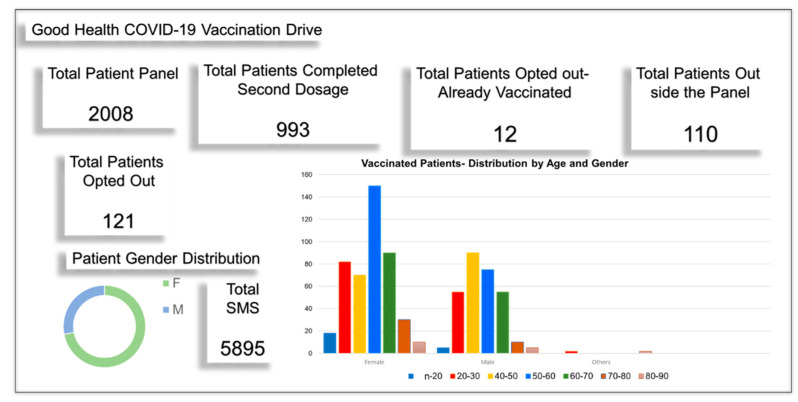
The GHAC achieved higher vaccination rates vs. surrounding Rutherford County.

**Figure 4 vaccines-10-01072-f004:**
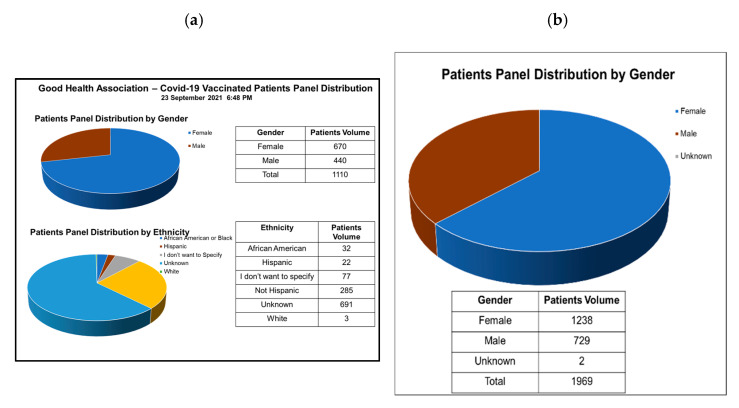
Patients panel distribution and vaccination: (**a**) by gender; (**b**) by ethnicity.

**Figure 5 vaccines-10-01072-f005:**
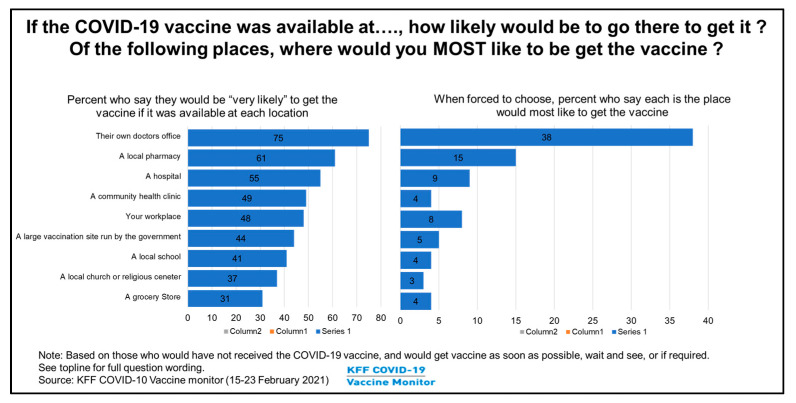
Vaccination location preferences of patients.

**Table 1 vaccines-10-01072-t001:** Binomial function for calculating probabilities for optimal utilization of vaccines.

Confidence level (Scheduled person will get the vaccine)	95%							
Probability scheduled person will come for vaccination (80%)	0.8							
Objective: Tom use 3 vials or 30 shots of vaccine	30							
n (number scheduled to receive vaccinations)	30	31	32	33	34	35	36	37
Number of Persons showing up for vaccination (X)	Probability (X)							
28	0.9895	0.9626	0.9069	0.8179	0.7004	0.5672	0.434	0.3141
29	0.9988	0.9913	0.9683	0.9192	0.8381	0.7279	0.5993	0.467
30	1	0.999	0.9929	0.9732	0.93	0.8565	0.7536	0.6302
31		1	0.9992	0.9941	0.9774	0.9395	0.8731	0.7775
32			1	0.9994	0.9952	0.981	0.9478	0.888
33				1	0.9995	0.996	0.984	0.955

**Note:** if 33 persons are scheduled then the probability that 30 or less will show up is 0.9732. In short, if we open 3 vials, we have 97.32% confidence that everyone that show up is vaccinated. If 34 persons are scheduled then the probability that 30 or less will show up is 0.93. In short, if we open 3 vials, we have 93% confidence that everyone that show up is vaccinated. This 93% confidence is less than 95% confidence level (One of our goals), therefore 33 persons scheduled swill provide the maximum utilization of vaccine minimizing dissatisfaction of not providing vaccine because clinic does not want to open new vial just for 1 or 2 persons.

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
