# Peer review of "COVID-19 Vaccination Drive in a Low-Volume Primary Care Clinic: Challenges & Lessons Learned in Using Homegrown Self-Scheduling Web-Based Mobile Platforms"

_vaccines, 2022, doi:10.3390/vaccines10071072_

Round 1

Reviewer 1 Report

This paper presents an interesting and useful work. Benefits and challenges of COVID-19 vaccine administration in a Primary Care Clinic in USA are described in detail, and opportunities for improvement in performance are described. Moreover, Primary Care Clinic administration advantages compared to administration in commercial pharmacies are shown.

It is a very detailed and worth publishing article.

I suggest to add, in the introduction, a section describing progress and challenges in the development of vaccines for COVID-19, including their safety profile and acceptance rates in several countries.

Figure 1 is not easy to read. It would be better to split into two figures and increase the size of characters.

Author Response

Vaccines-1743470

Title: COVID-19 vaccination drive in a Low Volume Primary Care Clinic: Challenges & Lessons learned in using Homegrown Self-scheduling web-based mobile platform

Reviewer #1

  1. This paper presents an interesting and useful work. Benefits and challenges of COVID-19 vaccine administration in a Primary Care Clinic in USA are described in detail, and opportunities for improvement in performance are described. Moreover, Primary Care Clinic administration advantages compared to administration in commercial pharmacies are shown. It is a very detailed and worth publishing article. I suggest to add, in the introduction, a section describing progress and challenges in the development of vaccines for COVID-19, including their safety profile and acceptance rates in several countries.

Response

Thanks for the positive support and constructive suggestion. Just to mention, the scenario for the COVID-19 vaccine is changing rapidly. The Moderna vaccine website has all the vaccine's updates regarding adverse events, age changes, and acceptance, or should I say, the spread of the vaccine, which will be available when that reference is opened and reviewed. It is reference#11. This article is to focus on the process of engaging patients as well as streamlining the process of vaccination to reduce the workload burden of our staff while playing our role in improving our town and state's vaccination spread. We just emphasized what was on CDC and positive media info on reducing hospitalization from covid and getting milder symptoms. We have added a section in the introduction describing progress and challenges in developing vaccines for COVID-19, including their safety profile and acceptance rates. Modifications are in Georgia Blue.

  1. Figure 1 is not easy to read. It would be better to split into two figures and increase the size of characters.

Response

Appreciate the suggestion to make this article better. We have split Figure 1 in two figures. They are now mentioned as Figure 1(a) and Figure 1(b) for easy read sake. Modifications are in Georgia Blue.

Reviewer 2 Report

This had low levels of novelty and interest to readers. This was just like report, not paper. Moreover, you need to edit your text as paper format without any grammatical errors.

Author Response

Vaccines-1743470

Title: COVID-19 vaccination drive in a Low Volume Primary Care Clinic: Challenges & Lessons learned in using Homegrown Self-scheduling web-based mobile platform

Reviewer #2

  1. This had low levels of novelty and interest to readers. This was just like report, not paper. Moreover, you need to edit your text as paper format without any grammatical errors.

Response:

We sincerely thank the reviewer for the suggestion regarding editing the paper for grammatical errors. We have thoroughly checked the article and revised it for grammatical mistakes with professional help. More than 700 corrections were made. For clarity, we have not changed the color for the changes. This has made the manuscript better. We also agree with the reviewer; this manuscript is now considered under "Case Report."

Reviewer 3 Report

This is useful paper with significant detail.   There are some small editorial changes required e.g. use of govt instead of the full term government.  

My main comment is providing much greater detail and background information about the PCC and their comparison communities.  GHCC is a small clinic of just over 200 patients and in comparing the vaccine take up they use two other clinics i.e. Rutherford County and Tennesse.   Maybe a small box would help other readers.  Really keen to understand if the differences are process or patient driven.

I would also suggest a key learnings section be added as there is significant amount of process detail in the paper that is difficult to fully understand. 

Finally what are next steps for this model with other vaccines e.g. Flu and future COVID vaccines. 

Author Response

Vaccines-1743470

Title: COVID-19 vaccination drive in a Low Volume Primary Care Clinic: Challenges & Lessons learned in using Homegrown Self-scheduling web-based mobile platform

Reviewer #3

  1. This is useful paper with significant detail.   There are some small editorial changes required e.g. use of govt instead of the full term government.

Response

We thank the reviewer for the positive comment. We have edited the paper with around 700 corrections; specifically, "govt" has been changed to "government." Modifications are in Georgia Blue (just for the government). We have thoroughly checked the paper and revised it for grammatical errors with professional help. More than 700 corrections were made. For clarity, we have not changed the color for the changes. This has made the manuscript better.

  1. My main comment is providing much greater detail and background information about the PCC and their comparison communities.  Really keen to understand if the differences are process or patient driven. I would also suggest a key learnings section be added as there is significant amount of process detail in the paper that is difficult to fully understand. Finally what are next steps for this model with other vaccines e.g. Flu and future COVID vaccines. 

Response

The comments from the reviewer are very reasonable and encouraging. Therefore, we have answered them here separately.

  • GHCC is a small clinic of just over 200 patients, and in comparing the vaccine take-up, they use two other clinics, i.e., Rutherford County and Tennessee.  
  • The paper is more process-driven for 2 things: focus on making it easier for patients(empower them to choose an appt) to have the appointment of their preference w/o going back and forth messaging. The second most important was to reduce the staff burden of enormous calls for setting up meetings and streamline the process to reduce documentation and just do the verification.
  • The process detail is given since, in standard clinical informatics, this is more a CDS (Clinical Decision System) 5-rights validation. Therefore, we gave a small paragraph as a summary before the process details were given.
  • In conclusion, we have given a summary of lessons learned.
  • Since Aug-Oct 2020, when we first wrote this paper, we have refined the mobile app and, as part of our website, keep refining how patients can sign up for appointments of their preference. Now patients signup for "reason of visit" and choose an appointment. We are hoping this way we can reduce our no-show times. We are still refining it.
  • Regarding other regular vaccines, now we just send a reminder about getting the vaccine, and if they have already received it, let us know. The 2nd part of the patient letting us know is still in the works. Again, the goal is to help us update our dashboard for insurance purposes to meet meaningful use and the population health data dashboard.